

# Estimation of reference evapotranspiration using some class-A pan evaporimeter pan coefficient estimation models in Mediterranean–Southeastern Anatolian transitional zone conditions of Turkey

Selçuk Usta

Construction Technology/Van Vocational School, Van Yüzüncü Yıl University, Van, Turkey

## ABSTRACT

**Background**. Reference evapotranspiration ($ET_o$), which is used as the basic data in many studies within the scope of hydrology, meteorology, irrigation and soil sciences, can be estimated by using the evaporation ($E_{pan}$) measured from the class-A pan evaporimeter. However, this method requires reliable pan coefficients ($K_p$). Many empirical models are used to estimate $K_p$ coefficients. The reliability of these models varies depending on climatic and environmental conditions. Therefore, they need to be tested in the local conditions where they will be used. In this study, conducted in Kahramanmaraş, which has a semi-arid Mediterranean climate in Turkey during the July–October periods of 2020 and 2021, aimed to determine the usability levels of six $K_p$ models in estimating daily and monthly average $ET_o$.

**Methods**. The $K_p$ coefficients estimated by the models were multiplied with the daily $E_{pan}$ values, and the daily average $ET_o$ values were estimated on the basis of the model. The daily $E_{pan}$ values were measured using an ultrasonic sensor sensitive to the water surface placed on the class-A pan evaporimeter. The ultrasonic sensor was managed by a programmable logic controller (PLC). To enable the sensor to be managed by PLC, a software was prepared using the CODESYS programming language and uploaded to the PLC. The daily average $ET_o$ values determined by the FAO-56 Penman–Monteith equation were accepted as actual values. The $ET_o$ values estimated by the $K_p$ models were compared with the actual $ET_o$ values using the mean absolute error (MAE), mean absolute percentage error (MAPE), root mean square error (RMSE) and determination coefficient ($R^2$) statistical approaches.

**Results**. The Wahed & Snyder outperformed the other models in estimating daily (MAE = 0.78 mm day$^{-1}$, MAPE = 14.40%, RMSE = 0.97 mm day$^{-1}$, $R^2$ = 0.82) and monthly (MAE = 0.32 mm day$^{-1}$, MAPE = 5.88%, RMSE = 0.32 mm day$^{-1}$, $R^2$ = 0.99) average $ET_o$. FAO-56 showed the nearest performance to Wahed & Snyder. The Snyder model presented the worst performance in estimating daily (MAE = 2.09 mm day$^{-1}$, MAPE = 37.53%, RMSE = 2.36 mm day$^{-1}$, $R^2$ = 0.82) and monthly (MAE = 1.83 mm day$^{-1}$, MAPE = 31.82%, RMSE = 1.87 mm day$^{-1}$, $R^2$ = 0.99) average $ET_o$. It has been concluded that none of the six $K_p$ models can be used to estimate the daily $ET_o$ in Kahramanmaraş located in the Mediterranean–Southeastern Anatolian transitional zone, and only Wahed & Snyder and FAO-56 can be used to estimate the monthly $ET_o$ without calibration.

Corresponding author
Selçuk Usta, susta@yyu.edu.tr

# INTRODUCTION

Evapotranspiration (ET) constitutes the most basic data for many studies such as determining the irrigation requirements of crops and preparing irrigation schedules, design, construction, and operation of irrigation–drainage systems, ponds, and dams, determining the amount of precipitation infiltrating into groundwater, and monitoring aridity (*Pandey, Dabral & Pandey, 2016*). ET can be most accurately measured using lysimeter systems. Installation and operational processes of these systems are complex and time consuming. Therefore, the approach of estimating ET by correcting $ET_o$ with the crop coefficient ($K_c$) is more preferred and widely used (*Şarlak & Bağçacı, 2020*).

Today, the most preferred method for estimating $ET_o$ is the Penman-Monteith. This method, created in 1948, was further developed by the Food and Agriculture Organization of the United Nations (FAO) in 1998 by adapting it to the grass reference crop and making it available under the name FAO-56 modification of the Penman-Monteith (PM) equation with Irrigation and Drainage Publication No. 56 (*Allen et al., 1998*). Numerous studies have revealed that the Penman-Monteith method is capable of estimating $ET_o$ values with high accuracy (*Lage et al., 2003*; *Jacobs et al., 2004*; *Trajković & Gocić, 2010*). As an alternative to the FAO-56 PM method, which is based on air temperature (T), relative humidity (RH), wind velocity at 2 m above ground surface ($U_2$), solar radiation ($R_s$), and soil heat flux (G), many empirical estimation methods based on T (*Thornthwaite, 1948*; *Blaney & Criddle, 1962*; *Hamon, 1961*), $R_s$ (*Makkink, 1957*; *Jensen & Haise, 1963*; *Priestley & Taylor, 1972*; *Doorenbos & Pruitt, 1977*), both T and $R_s$ (*Turc, 1961*; *Hargreaves & Samani, 1985*) have been developed. The climate data needed for both FAO-56 PM and other empirical estimation methods are measured by meteorological ground observation stations. Although these stations are not widespread enough around the world, they are mostly located in city centres. Therefore, climate data cannot be measured continuously and regularly in rural areas. This situation limits the usability of empirical $ET_o$ estimation methods such as FAO-56 PM, Thornthwaite (*Thornthwaite, 1948*), Blaney & Criddle (*Blaney & Criddle, 1962*), Makkink (*Makkink, 1957*), Jensen & Haise (*Jensen & Haise, 1963*), Priestley & Taylor (*Priestley & Taylor, 1972*), Turc (*Turc, 1961*), and Hargreaves & Samani (*Hargreaves & Samani, 1985*) (*El-Sebaii et al., 2010*).

Unlike the methods of lysimeter and empirical estimation, in the class-A pan evaporimeter method, the $E_{pan}$ from the water surface is corrected by the $K_p$ coefficient and $ET_o$ can be estimated depending on only one parameter. Reliable $K_p$ coefficients are needed in this method, which is widely preferred in $ET_o$ estimation due to the low-cost and simplicity of the technique used. To determine $K_p$ coefficients, many estimation models were developed as a function of the upwind buffer zone distance (FET), $U_2$, and RH around the class-A pan evaporimeter (*Cuenca, 1989*; *Snyder, 1992*; *Abdel-Wahed & Snyder, 2008*; *Allen et al., 1998*; *Grismer et al., 2002*; *Orang, 1998*; *Pereira et al., 1995*; *Raghuwanshi*

& Wallender, 1998). However, since these methods are compatible with the climatic and environmental characteristics of the region, where they were developed, their reliability should be tested if they are used in different regions (*Jensen, Burman & Allen, 1990; Irmak, Haman & Jones, 2002*). Numerous studies have been conducted in many regions with diverse climatic and environmental characteristics. In these studies, $ET_o$ values obtained by $K_p$ estimation models were compared with $ET_o$ values determined using the lysimeter or empirical estimation models. *Sentelhas & Folegatti (2003)* estimated $ET_o$ values using some $K_p$ coefficient estimation models for a semi-arid region in Brazil and compared these values with actual $ET_o$ values measured by a weighing lysimeter. They indicated that the Pereira and Cuenca models were the best for estimating $ET_o$. *Gundekar et al. (2008)*, *Kaya et al. (2012)*, and *Pradhan et al. (2013)* reported that Snyder and Pereira were the models with the best and worst estimating performances, respectively, in the semi-arid conditions. *Aydın (2019)* declared that the Snyder model performed better than the Pereira model in the semi-arid Southeastern Anatolia region of Turkey. *Kumar Kar et al. (2017)* estimated the $ET_o$ values nearest to the $ET_o$ values obtained by the FAO-56 PM equation using the Orang model in a study conducted in semi-arid conditions of Nigeria. *Sabziparvar et al. (2010)* reported that Snyder was the model that performs best in Iran's warm-arid climate. *Irmak, Haman & Jones (2002)*; *SreeMaheswari & Jyothy (2017)*; *Kar et al. (2017)*; *Khobragade et al. (2019)*, and *Mahmud et al. (2020)* revealed that Snyder and Cuenca are the models with the highest estimating performance in their studies conducted in humid regions of the United States of America, India and Bangladesh, respectively. *Rodrigues et al. (2020)* developed a new model based on T, RH, $R_s$, and $U_2$ parameters in Portuguese conditions with a Mediterranean climate. They obtained determination coefficients ($R^2$) ranging from 0.67 to 0.74 as an expression of the statistical relationship between the $ET_o$ values estimated with this model and the $ET_o$ values determined using the Eddy covariance method. *Aschonitis, Antonopoulos & Papamichail (2012)* concluded that the models with the best and worst estimating performances were Cuenca and Snyder, respectively, in their study realised in the Thessaloniki Plain of Greece, which has a semi-arid Mediterranean climate. *Koç (2022)* stated that in Adana, located in southern Turkey with a hot-summer Mediterranean climate, the models with the best and worst estimating performances were Wahed & Snyder and Snyder, respectively. It has been observed that the reliability and usability levels of the $K_p$ coefficient estimation models evaluated within the scope of these studies vary depending on climatic and environmental conditions. Reliable $K_p$ coefficients are needed for daily average $ET_o$ estimates based on $E_{pan}$ measured from the class-A pan evaporimeter. Therefore, $K_p$ coefficient estimation models need to be tested in the local conditions where they will be used and calibrated if necessary.

Kahramanmaraş is one of the cities with high agricultural production potential in Turkey. In this city, which has the climatic and environmental characteristics of both the Mediterranean and Southeastern Anatolia regions, crop production activities are conducted mostly in rural areas. Many of the climate parameters required for the FAO-56 PM method, which is the most preferred method in irrigation activities based on crop water consumption, cannot be measured continuously and regularly in these rural areas.

Therefore, there is a need to use low-cost methods, which have simple usage techniques, such as a class-A pan evaporimeter.

This study conducted in Kahramanmaraş with a Mediterranean climate, aimed to compare the Cuenca, Snyder, Wahed & Snyder, FAO-56, Modified Snyder, and Orang $K_p$ coefficient estimation models, and to determine their usability levels in estimating daily average $ET_o$. It was targeted to determine the most appropriate $K_p$ coefficient estimation models that can be used in $E_{pan}$-based daily average $ET_o$ estimates in Kahramanmaraş, located in the Mediterranean-Southeastern Anatolia transitional zone of Turkey.

## MATERIALS & METHODS

Kahramanmaraş is located between 37°11′–38°26′ north latitudes and 36°15′–37°42′ east longitudes in the Mediterranean-Southeastern Anatolian transitional zone of Turkey, and its altitude is 568 m (Fig. 1). The annual averages of the air temperature and relative humidity are 16.90 °C and 58.34%, respectively. In parts of the city with an altitude of up to 1,000 m, the Mediterranean climate is dominant, with hot and dry summers and mild and rainy winters. In parts with an altitude of more than 1,000 m, the effects of the Mediterranean mountain climate are felt, with cold and snowy winters and relatively cool summers. Kahramanmaraş, with a annual total precipitation of 721.60 mm, is located in the semi-arid climatic zone. During the May–October period, when the daily maximum air temperature varying between 26.10–36.10 °C, precipitation decreases considerably. In this period, the monthly total precipitation varying between 2.20–45.40 mm is insufficient to satisfy the crop water consumption and irrigation becomes mandatory (*Turkish State Meteorological Service, 2022*).

This study was conducted in the research field established on the Kahramanmaraş Sütçü İmam University campus, July–October periods of 2020 and 2021. The research field is located at 37°35′36″ north latitude and 36°49′20″ east longitude, with an altitude of 508 m.

Firstly, the daily average $ET_o$ values were determined by using the FAO-56 PM Eq. (1). These values were accepted as actual $ET_o$ values. The components of Eq. (1) were determined using the Irrigation and Drainage Publication No. 56 (*Allen et al., 1998*).

$$ET_o = \frac{0.408\Delta(R_n - G) + \gamma\left(\frac{900}{T+273}\right)U_2(e_s - e_a)}{\Delta + \gamma(1 + 0.34U_2)} \tag{1}$$

where $ET_o$ = reference evapotranspiration (mm day$^{-1}$); $\Delta$ = slope of saturation vapour pressure curve (kPa/°C$^{-1}$); $R_n$ = net radiation (MJ m$^{-2}$ day$^{-1}$); G = soil heat flux (MJ m$^{-2}$ day$^{-1}$); $\gamma$ = psychrometric constant (kPa/° C$^{-1}$); $e_s$ = saturation vapour pressure (kPa); $e_a$ = actual vapour pressure (kPa); $e_s$ –$e_a$ = vapour pressure deficit (kPa); $U_2$ = wind velocity at 2 m above ground surface (m s$^{-1}$); T = daily average air temperature (°C) (*Allen et al., 1998*).

Secondly, by measuring the daily $E_{pan}$ values from the class-A pan evaporimeter installed in the research field, the daily actual $K_p$ coefficients were determined by Eq. (2) (*Doorenbos*

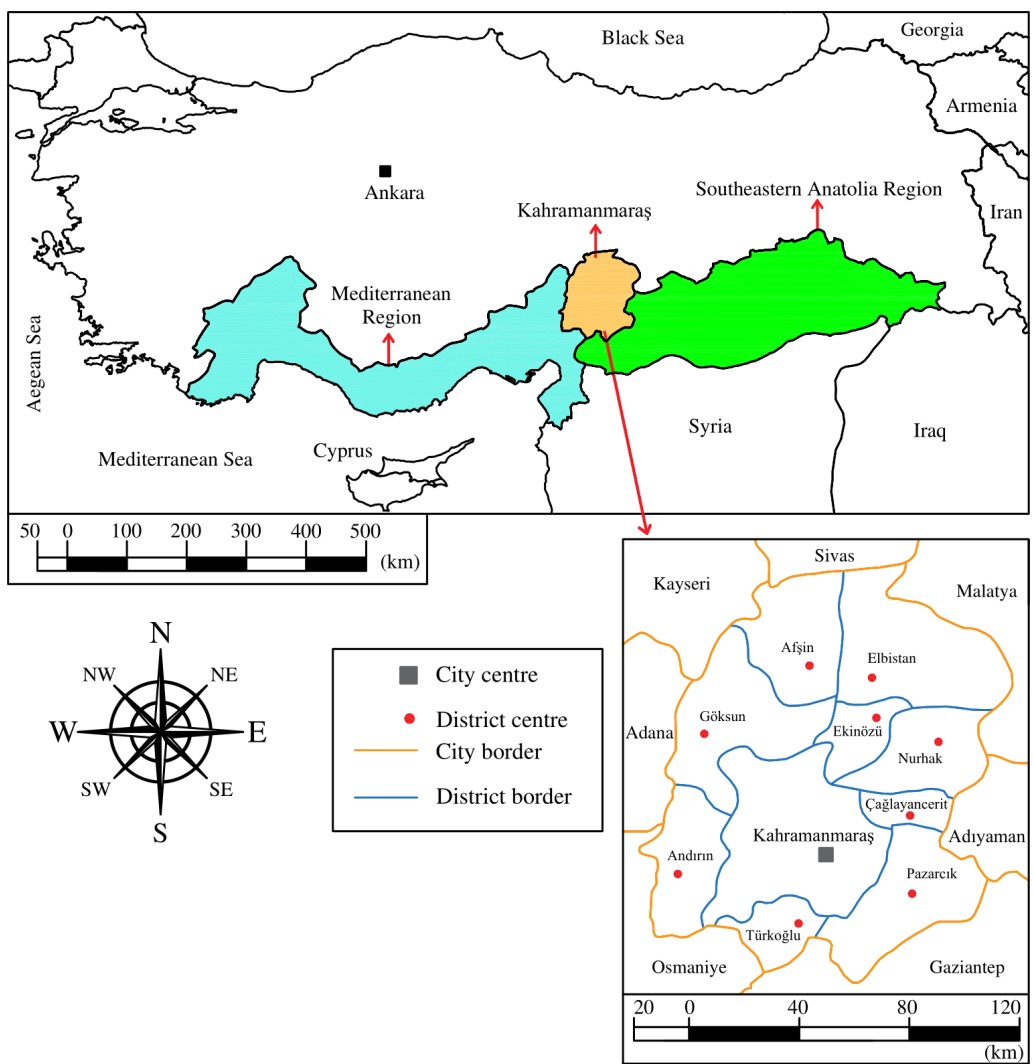

**Figure 1  Geographical location of Kahramanmaraş in Turkey map.** Drawing credit: Selçuk Usta.

*& Pruitt, 1977*; *Allen et al., 1998*).

$$ET_o = E_{pan} . K_p \qquad K_p = \frac{ET_o}{E_{pan}} \qquad (2)$$

where $E_{pan}$ = pan evaporation (mm day$^{-1}$); $K_p$ = pan coefficient.

Thirdly, the $K_p$ coefficients were estimated using the models of Cuenca (*Cuenca, 1989*), Snyder (*Snyder, 1992*; *Abdel-Wahed & Snyder, 2008*), FAO-56 (*Allen et al., 1998*), Modified Snyder (*Grismer et al., 2002*) and Orang (*Orang, 1998*). These models developed as a function of the FET, $U_2$ and RH around the Class-A pan evaporimeter are given in Table 1. The evaporimeter used in this study was placed on dry fallow soil surrounded by green crops at an average distance of 20 m. For this reason, the FET distance was considered as 20 m.

**Table 1  Class-A pan coefficient estimation models.**

| Model | Estimation equation |
|---|---|
| Cuenca | $K_p = 0.475 - 0.00024U_2 + 0.00516RH + 0.00118(FET) - 0.000016(RH)^2 - 0.00000101(FET)^2 - 0.000000008(RH)^2U_2 - 0.00000001(RH)^2(FET)$ |
| Snyder | $K_p = 0.482 - 0.000376U_2 + 0.0424Ln(FET) + 0.0045RH$ |
| Wahed & Snyder | $K_p = 0.62407 - 0.00028U_2 - 0.02660Ln(FET) + 0.00226RH$ |
| FAO-56 | $K_p = 0.61 + 0.000162U_2RH - 0.00000959U_2(FET) + 0.00341RH + 0.00327U_2Ln(FET) - 0.00289U_2Ln(86.4U_2) - 0.0106Ln(86.4U_2)Ln(FET) + 0.00063[Ln(FET)]^2Ln(86.4U_2)$ |
| Modified Snyder | $K_p = 0.5321 - 0.0003U_2 + 0.0249Ln(FET) + 0.0025RH$ |
| Orang | $K_p = 0.51206 - 0.000321U_2 + 0.03188Ln(FET) + 0.00289RH - 0.000107RH\,Ln(FET)$ |

**Notes.**

$K_p$, pan coefficient; $U_2$, wind velocity at 2 m above ground surface (m s$^{-1}$); RH, relative humidity (%); FET, class-A pan evaporimeter upwind buffer zone distance (m).

Finally, the $K_p$ coefficients determined using the models were multiplied by the daily $E_{pan}$ values, and the daily $ET_o$ values were estimated on the basis of the model. The estimated $ET_o$ values were compared with the actual $ET_o$ values determined by the FAO-56 PM equation. Thus, the accuracy and reliability levels of the pan coefficient estimation models have been revealed.

Daily T, RH, $U_2$ and $R_s$ used as input variables in the FAO-56 PM and $K_p$ estimation models were measured at the climate station given in Fig. 2. The sensors on the climate station have been managed by the PM 590 PLC.

PM 590 PLC has an SD card with 2 GB memory, 160 analog inputs, 160 analog outputs, 320 digital inputs and 240 digital outputs. It generates numerical values (NV) varying between 1–27648 for input signals varying between of 4–20 mA or 0–10 V (ABB, 2020a). The temperature and humidity sensors can measure with an accuracy of $\pm0.20$ °C and $\pm2.50$% in the ranges of 0–70 °C and 10–90%, respectively. Similarly, solar radiation and wind velocity sensors can measure with an accuracy of 7.00 μV Watt$^{-1}$ m$^{-2}$ and 0.10 m s$^{-1}$ in the ranges of 0–2,000 W m$^{-2}$ and 0.40–30 m s$^{-1}$, respectively (ONSET, 2020; EKO, 2020; NESA, 2020a; NESA, 2020b). To enable the sensors to be managed by PLC, software was prepared using the CODESYS programming language and uploaded to the PLC (ABB, 2020b). This software measured the air temperature and relative humidity every hour on the hour, solar radiation and wind velocity every half hour during one-day periods and recorded them on the SD card on the PLC. The 24-hour period between 08:59:30 on the previous day and 08:59:30 on the next day was taken into account as a one-day period.

The temperature and humidity sensors generate output signals varying between of 4–20 mA for the values of varying between of 0–100 °C and 0–100%, respectively. These signals were firstly converted to numerical values varying between 0 to 27,648 by the PLC, and then to the values of hourly temperature in °C (Eq. 3a) and hourly humidity in % (Eq. 3b) by the software. The numerical value generated by the PLC for the maximum values of temperature (100 °C) and humidity (100%) is 27,648. The software obtained the daily maximum and minimum values of air temperature and relative humidity by sorting the hourly temperature and humidity data, from the biggest to the smallest, at the end of the

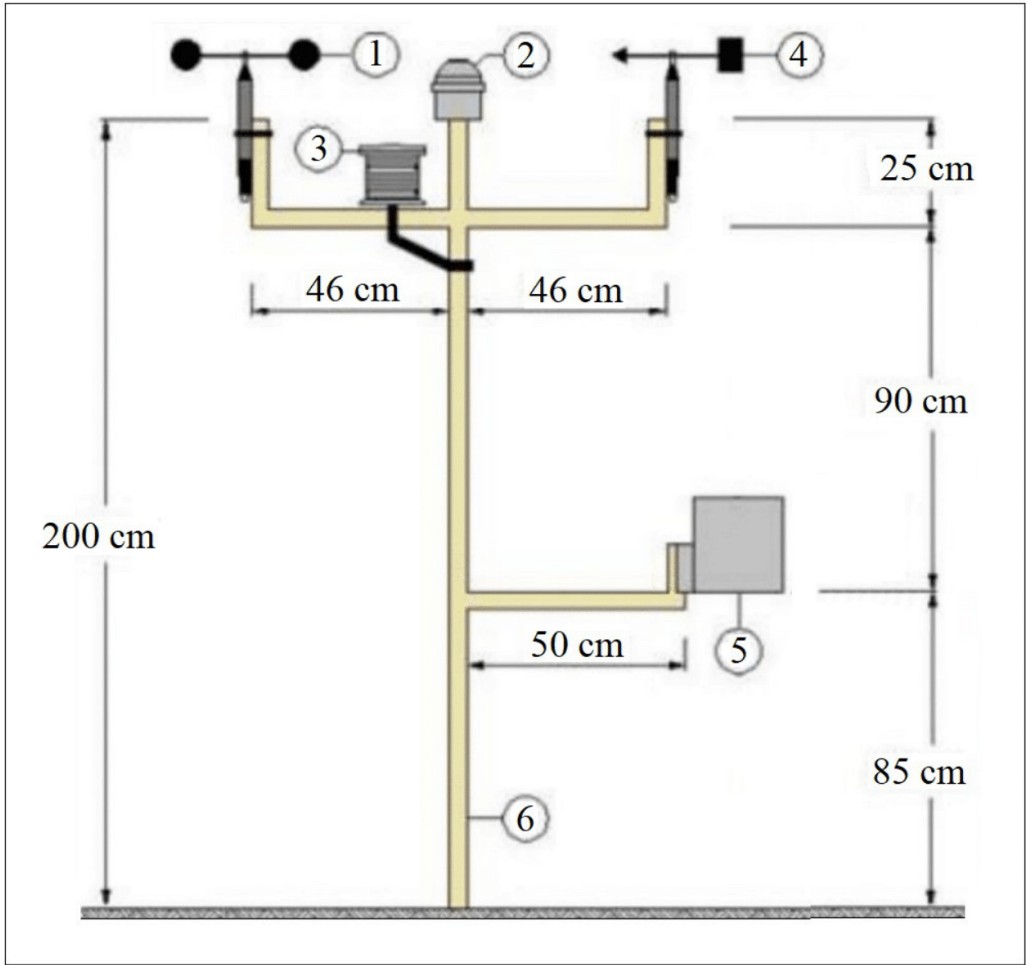

**Figure 2  Programmable logic controller (PLC) controlled climate station.** This station consists of sensors wind velocity (1), solar radiation (2), air temperature–relative humidity (3), wind direction (4) and precipitation (5). These sensors were mounted on a platform (6) made of steel pipe profile. Drawing credit: Selçuk Usta.

day. Then, it determined the daily average air temperature (Eq. 4a) and relative humidity (Eq. 4b) by calculating the averages of daily maximum and minimum air temperature and relative humidity values.

$$T_h = \frac{NV.100}{27648} \quad (a) \qquad RH_h = \frac{NV.100}{27648} \quad (b) \tag{3}$$

$$T = \frac{T_{max} + T_{min}}{2} \quad (a) \qquad RH = \frac{RH_{max} + RH_{min}}{2} \quad (b) \tag{4}$$

where $T_h$ = hourly air temperature (°C); NV = numerical value generated by PLC (0–27648); $RH_h$ = hourly relative humidity (%); $T_{max}$ = daily maximum air temperature (°C); $T_{min}$ = daily minimum air temperature (°C); $RH_{max}$ = daily maximum relative humidity (%); $RH_{min}$ = daily minimum relative humidity (%); T = daily average air temperature (°C); RH = daily average relative humidity (%).

The solar radiation and wind velocity sensors generate output signals varying between of 0–10 V for the values of varying between of 0–2,000 W m$^{-2}$ and 0.28–50 m s$^{-1}$, respectively. The signals generates by the radiation sensor were firstly converted to numerical values varying between 0 to 27,648 by the PLC, and then to the half-hourly solar radiation values by the software (Eq. 5a). Similarly, the signals generates by the wind velocity sensor were firstly converted to numerical values varying between 0 to 5,530 by the PLC, and then to the half-hourly wind velocity values by the software (Eq. 5b). The numerical values generated by the PLC for the maximum values of the solar radiation (2,000 W m$^{-2}$) and wind velocity (50 m s$^{-1}$) are 27,648 and 5,530, respectively.

$$RS_{h/2} = \frac{NV.2000}{27648} \quad (a) \qquad U_{h/2} = \frac{NV.50}{5530} \quad (b) \tag{5}$$

Where $RS_{h/2}$ = half-hourly solar radiation (Watt m$^{-2}$); $U_{h/2}$ = half-hourly wind velocity (m s$^{-1}$).

The software summed the half-hourly solar radiation and wind velocity data at the end of the day, and obtained the daily total values of the solar radiation and wind velocity. The daily total values were divided by the number of measurements (48) by the software and determined the daily average solar radiation (Eq. 6a) and wind velocity (Eq. 6b). The solar radiation sensor measures in Watt m$^{-2}$ unit. However, solar radiation is used in unit of MJ m$^{-2}$ day$^{-1}$ in the FAO-56 PM equation. For this reason, the values measured in Watt m$^{-2}$ unit were multiplied by the coefficient of 0.0864 and converted to MJ m$^{-2}$ day$^{-1}$ unit.

$$R_s = \left( \frac{\sum RS_{h/2}}{48} \right) 0.0864 \quad (a) \qquad U_2 = \frac{\sum U_{h/2}}{48} \quad (b) \tag{6}$$

where $\sum RS_{h/2}$ = daily total solar radiation (MJ m$^{-2}$ day$^{-1}$); $\sum U_{h/2}$ = daily total wind velocity (m s$^{-1}$); $R_s$ = daily average solar radiation (MJ m$^{-2}$ day$^{-1}$); $U_2$ = daily average wind velocity (m s$^{-1}$).

Daily $E_{pan}$ values were measured using an ultrasonic sensor sensitive to the water surface placed on the class-A pan evaporimeter given in Fig. 3.

To enable the ultrasonic and pressure sensors and solenoid valve to be managed by PLC, software was prepared using the CODESYS programming language and uploaded to the PLC (*ABB, 2020b*). This software performed the measurements for one-day periods. The 24-hour period between 08:59:30 on the previous day and 08:59:30 on the next day was considered as a one-day period. The ultrasonic sensor generates output signals varying between 4–20 mA for distances varying between 0–500 mm (*Pepperl+Fuchs Group, 2020*). These signals generated by the sensor for the height (0–500 mm) between itself and the water surface were firstly converted to numerical values varying between 0 to 27,648 by the PLC, and then to the actual height distance values in mm by the software (Eq. 7). The numerical value generated by the PLC for the maximum height ($H = 500$ mm) is 27,648. Finally, the software determined the water level in the Class-A pan evaporimeter by using Eq. (8) and recorded it on the SD card. Daily $E_{pan}$ was determined by subtracting the water levels measured at the beginning and end of a one-day period (Eq. 9). Measuring the water level in the evaporimeter was started when the water level was 200 mm. When the water

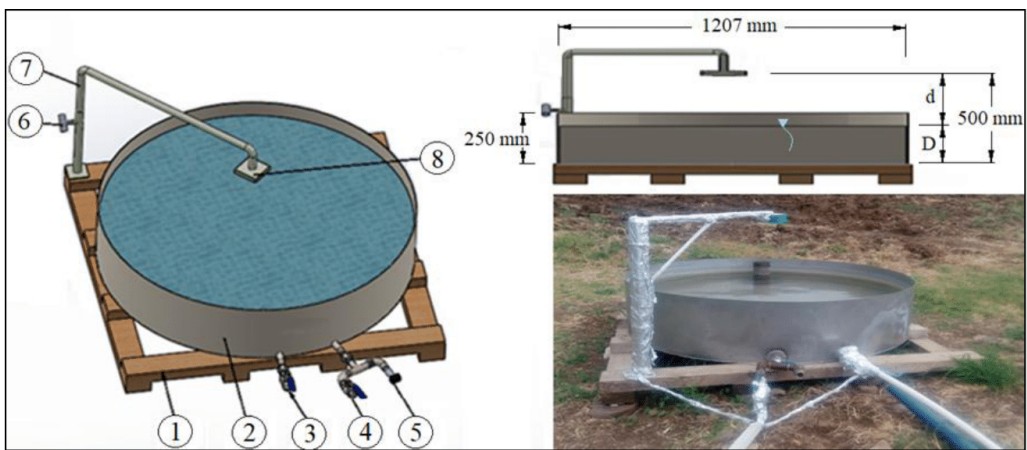

**Figure 3  Programmable logic controller (PLC) controlled class-A pan evaporimeter.** This evaporimeter (2) was sited on a 10 cm high wooden frame (1) placed on dry fallow soil surrounded by green crops. The pipes of the water inlet (3) and discharge (4) were placed on the bottom of the evaporimeter. Both of these pipes have a diameter of $\frac{1}{2}$". A solenoid valve was connected to the water inlet pipe. The $E_{pan}$ values can be measured separately by using a pressure sensor (5) placed on the discharge pipe or an ultrasonic sensor (8) sensitive to the water surface. The $E_{pan}$ values measured by the ultrasonic sensor were used in this study. This sensor was placed at a height of 500 mm, coinciding with the centre of the evaporimeter, by means of a strut (7) with a height adjustment screw (6) on it. Drawings credit: Selçuk Usta; photo credit: Selçuk Usta (Gençoğlan, Gençoğlan, and Usta, 2023); CC BY NC 4.0.

level falls below 150 mm, the PLC opens the solenoid valve, allowing water to be supplied to the evaporimeter until the water level reaches 200 mm. The valve is automatically closed by the PLC, when the water level reaches 200 mm.

$$d = \frac{NV.500}{27648} \qquad (7)$$

$$D = 500 - d \qquad (8)$$

$$E_{pan} = D_{beg.} - D_{end} \qquad (9)$$

where d = the height distance between the ultrasonic sensor and the water surface (mm); D = the water level in the pan evaporimeter (mm); $D_{beg.}$ = the water level measured at the beginning of a one-day period (mm); $D_{end}$ = the water level measured at the end of a one-day period (mm).

The daily average actual and estimated $ET_o$ values were compared using the statistical approaches of the mean absolute error, mean absolute percentage error, and root mean square error. These errors were determined using Eqs. (10)–(12), respectively. Mean absolute percentage error was taken into account in revealing the accuracy levels of the $ET_o$ values estimated using the daily average $K_p$ coefficients determined by the models. The accuracy of the estimated $ET_o$ values; mean absolute percentage error was evaluated as "excellent" if it was less than 10%, "good" if it was between 10–20%, "reasonable" if it was between 20–50%, and "inaccurate" if it was more than 50% (*Lewis, 1982*). Regression analyses were performed using Microsoft Excel software to reveal the level of statistical

relationship between actual and estimated daily average $ET_o$ values and the results were discussed (Eq. 13).

$$MAE = \frac{1}{n}\sum_{i:1}^{n}|X_i - Y_i| \qquad (10)$$

$$MAPE = \frac{1}{n}\sum_{i:1}^{n}\left|\frac{X_i - Y_i}{X_i}\right| \times 100 \qquad (11)$$

$$RMSE = \sqrt{\frac{1}{n}\sum_{i:1}^{n}(X_i - Y_i)^2} \qquad (12)$$

$$R^2 = \frac{\left[\sum_{i:1}^{n}(X_i - \hat{X})(Y_i - \hat{Y})\right]^2}{\sum_{i=1}^{n}(X_i - \hat{X})^2 \sum_{i=1}^{n}(Y_i - \hat{Y})^2} \qquad (13)$$

where MAE = mean absolute error (mm day$^{-1}$); MAPE = mean absolute percentage error (%); RMSE = root mean square error (mm day$^{-1}$); $X_i$ and $Y_i$ = actual and estimated $ET_o$ values (mm day$^{-1}$); $\hat{X}$ and $\hat{Y}$ = averages of the actual and estimated $ET_o$ values (mm day$^{-1}$); $R^2$ = determination coefficient; n = number of observations (123 days).

## RESULTS

The daily average air temperature, solar radiation and wind velocity values measured during the July–October periods of 2020 and 2021 generally showed a decreasing trend. Relative humidity values exhibited an increasing trend during the same periods. Air temperature, solar radiation, wind velocity and relative humidity values ranged between 17.66–30.10 °C, 10.51–30.23 MJ m$^{-2}$ day$^{-1}$, 0.40–4.23 m s$^{-1}$ and 24.50–61.30%, respectively, in the first year. The same values ranged between 17.66–30.10 °C, 10.40–29.23 MJ m$^{-2}$ day$^{-1}$, 0.43–4.65 m s$^{-1}$ and 30.20–67.80%, respectively, in the second year (Figs. 4 and 5).

The daily average actual $ET_o$ values determined using the FAO-56 PM equation varied between 2.20–8.93 mm day$^{-1}$ and 1.77–9.60 mm day$^{-1}$ in the July–October periods of 2020 and 2021, respectively. The daily total $E_{pan}$ values measured from the class-A pan evaporimeter varied between 3.00–16.00 mm day$^{-1}$ and 3.00–15.00 mm day$^{-1}$, respectively, in the same periods (Fig. 6). The daily $ET_o$ and $E_{pan}$ values generally showed a decreasing trend during the July–October periods of 2020 and 2021. These values were increased to maximum levels in the last period of July and the first and second periods of August. It has been observed that the daily $ET_o$ and $E_{pan}$ values realised on the days when the air temperature, wind velocity, and solar radiation were at high levels and the relative humidity was at low levels, were higher than the other days. As an expression of the correlation between daily $ET_o$ and $E_{pan}$ values, $R^2$ coefficients were determined as 0.83 and 0.78 for the July–October periods of 2020 and 2021, respectively.

The daily actual $K_p$ coefficients obtained using the daily average $ET_o$ values determined by the FAO-56 PM equation and the daily total $E_{pan}$ values measured from the class-A pan evaporimeter ranged between 0.38–0.88 in the first year and 0.35–1.08 in the second year. Seasonal average coefficients were determined as 0.60 and 0.65, respectively. Similarly,

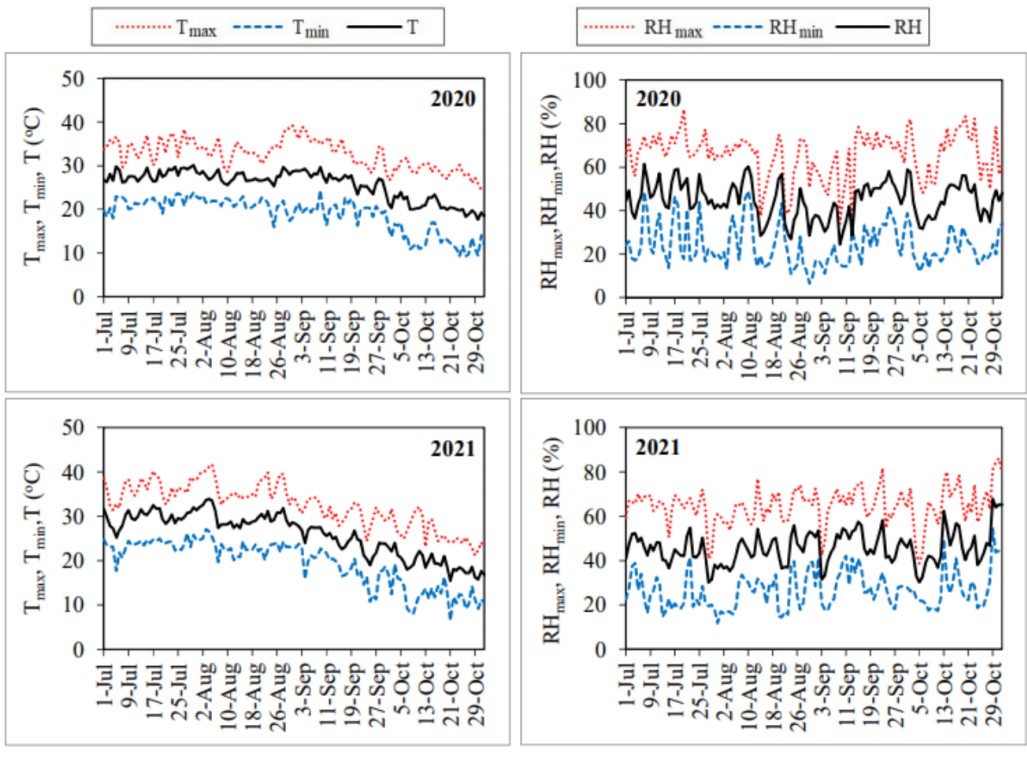

**Figure 4** Daily air temperature ($T_{max}$, $T_{min}$, T) and relative humidity ($RH_{max}$, $RH_{min}$, RH) values for the July–October periods of 2020 and 2021.

**Figure 5** Daily average wind velocity ($U_2$) and solar radiation ($R_s$) values. Each point on the graphs represents the daily average $ET_o$ and $R_s$ values for the July–October periods of 2020 and 2021.

the daily $K_p$ coefficients estimated using the Cuenca, FAO-56, Modified Snyder, Orang, Snyder and Wahed & Snyder models for both years varied between 0.61–0.77, 0.52–0.71, 0.67–0.78, 0.67–0.78, 0.72–0.91, and 0.60–0.70, respectively. Seasonal average coefficients

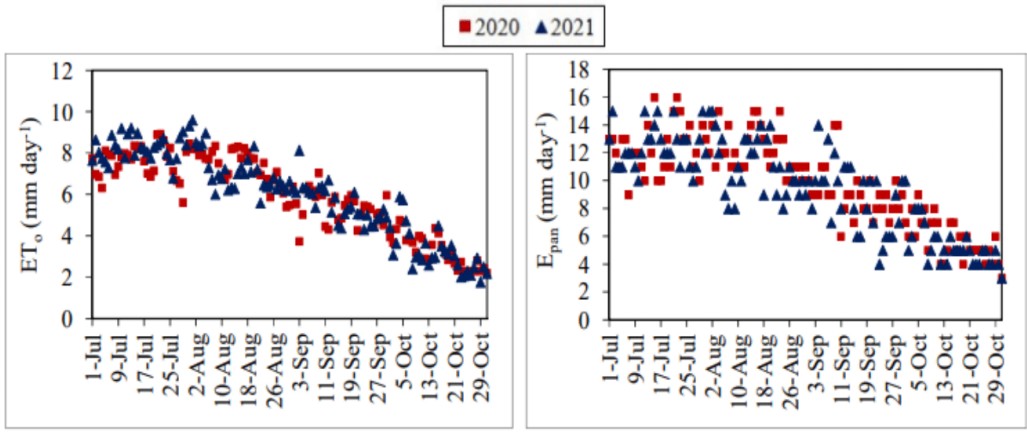

**Figure 6 Daily average actual reference evapotranspiration (ET$_o$) and daily total pan evaporation (E$_{pan}$) values.** Each point on the graphs represents the daily actual ET$_o$ and E$_{pan}$ values for the July–October periods of 2020 and 2021.

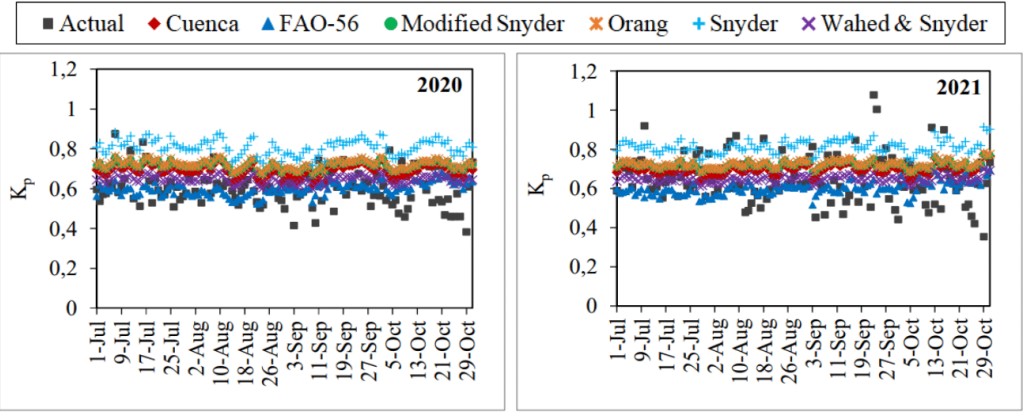

**Figure 7 Daily average actual and estimated pan coefficients (K$_p$).** Each point on the graphs represents the daily K$_p$ values for the July–October periods of 2020 and 2021.

were determined as 0.70, 0.60, 0.62, 0.72, 0.81 and 0.65 respectively. It has been observed that the K$_p$ coefficients estimated using the models of Modified Snyder and Orang were very similar to each other (Fig. 7).

The monthly average actual K$_p$ coefficients were determined as 0.62 for July, 0.60 for August, 0.61 for September and 0.58 for October in the first year. The same coefficients were obtained for the second year as 0.67, 0.65, 0.67 and 0.61, respectively. The nearest values to the actual coefficients were estimated by the FAO-56 (0.57–0.63) in the first year and by the Wahed & Snyder (0.64–0.65) in the second year. The furthest values were estimated by the Snyder (0.80–0.82) in both years. Generally, it has been observed that the K$_p$ coefficient changes directly proportional to the humidity, which tends to increase

**Table 2** Monthly averages of the actual and estimated daily $K_p$ coefficients.

| Model/Month (2020) | July | August | September | October | Average |
|---|---|---|---|---|---|
| Actual | 0.62 | 0.60 | 0.61 | 0.58 | 0.60 |
| Cuenca | 0.71 | 0.69 | 0.69 | 0.69 | 0.70 |
| Snyder | 0.82 | 0.80 | 0.80 | 0.81 | 0.81 |
| Wahed & Snyder | 0.65 | 0.64 | 0.64 | 0.65 | 0.65 |
| FAO-56 | 0.59 | 0.57 | 0.60 | 0.63 | 0.60 |
| Modified Snyder | 0.73 | 0.71 | 0.72 | 0.72 | 0.72 |
| Orang | 0.73 | 0.72 | 0.72 | 0.72 | 0.72 |
| **Model/Month (2021)** | **July** | **August** | **September** | **October** | **Average** |
| Actual | 0.67 | 0.65 | 0.67 | 0.61 | 0.65 |
| Cuenca | 0.69 | 0.70 | 0.71 | 0.70 | 0.70 |
| Snyder | 0.81 | 0.81 | 0.82 | 0.82 | 0.82 |
| Wahed & Snyder | 0.64 | 0.65 | 0.65 | 0.65 | 0.65 |
| FAO-56 | 0.58 | 0.59 | 0.60 | 0.63 | 0.60 |
| Modified Snyder | 0.72 | 0.72 | 0.72 | 0.72 | 0.72 |
| Orang | 0.72 | 0.72 | 0.73 | 0.73 | 0.73 |

during the July–October period, and inversely proportional to the wind velocity, which tends to decrease in the same period (Table 2).

The daily average $ET_o$ values estimated using the $K_p$ coefficients determined with the Cuenca, FAO-56, Modified Snyder, Orang, Snyder and Wahed & Snyder models varied between 2.09–10.97 mm day$^{-1}$, 1.91–9.15 mm day$^{-1}$, 2.15–11.34 mm day$^{-1}$, 2.16–11.40 mm day$^{-1}$, 2.43–12.82 mm day$^{-1}$ and 1.93–10.18 mm day$^{-1}$ in the first year, respectively. The seasonal average values were determined as 6.83 mm day$^{-1}$, 5.83 mm day$^{-1}$, 7.07 mm day$^{-1}$, 7.10 mm day$^{-1}$, 7.96 mm day$^{-1}$ and 6.35 mm day$^{-1}$, respectively. In the same year, the daily average actual $ET_o$ values varied between 2.20–8.93 mm day$^{-1}$. The seasonal average actual $ET_o$ was determined as 5.91 mm day$^{-1}$. The nearest values to the actual $ET_o$ values were estimated by the FAO-56, and the furthest values were estimated with the Snyder in the first year. Except for the FAO-56, the nearest values to the actual $ET_o$ values were obtained by using the models of Wahed & Snyder, Cuenca, Modified Snyder, Orang and Snyder, respectively (Fig. 8). The daily average $ET_o$ values estimated using the Cuenca, FAO-56, Modified Snyder, Orang, Snyder and Wahed & Snyder models varied between 2.30–10.80 mm day$^{-1}$, 2.08–8.70 mm day$^{-1}$, 2.31–11.01 mm day$^{-1}$, 2.32–11.07 mm day$^{-1}$, 2.71–12.57 mm day$^{-1}$ and 2.07–9.89 mm day$^{-1}$ in the second year, respectively. The seasonal average values were determined as 6.56 mm day$^{-1}$, 5.57 mm day$^{-1}$, 6.77 mm day$^{-1}$, 6.80 mm day$^{-1}$, 7.63 mm day$^{-1}$ and 6.08 mm day$^{-1}$, respectively. In the same year, the daily average actual $ET_o$ values ranged between 1.77–9.60 mm day$^{-1}$. The seasonal average actual $ET_o$ was determined as 6.03 mm day$^{-1}$. Unlike the first year, the nearest values to the actual $ET_o$ values were estimated by Wahed & Snyder in the second year. The furthest values were estimated with the Snyder as in the first year. Except for the Wahed & Snyder in the second year, the nearest values to the actual $ET_o$ values were estimated by

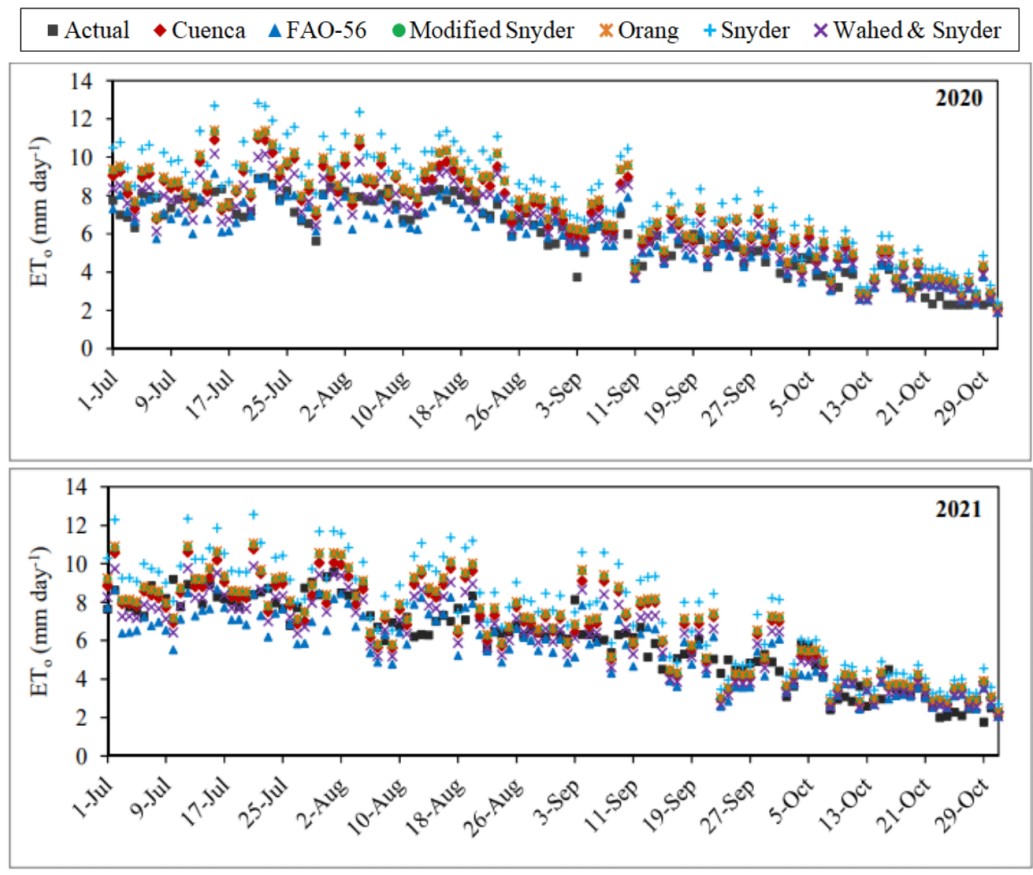

**Figure 8  Daily average actual and estimated reference evapotranspiration (ET$_o$) values.** Each point on the graphs represents the actual and estimated ET$_o$ values for the July–October periods of 2020 and 2021.

using the models of FAO-56, Cuenca, Modified Snyder, Orang, and Snyder, respectively, as in the first year (Fig. 8).

The daily average ET$_o$ values with the lowest and highest errors in the first year were estimated using the FAO-56 and Snyder models, respectively. The MAE, MAPE, and RMSE errors determined for the FAO-56 model, which has the best-estimating performance, varied between 0.56–0.68 mm day$^{-1}$, 8.79–18.78% and 0.66–0.93 mm day$^{-1}$, respectively. Seasonal average errors for the July–October period were realised as 0.62 mm day$^{-1}$, 11.81% and 0.79 mm day$^{-1}$, respectively. The MAE, MAPE and RMSE errors obtained for the Snyder model, which has the worst estimating performance, varied between 1.53–2.81 mm day$^{-1}$, 36.42–49.06% and 1.73–2.99 mm day$^{-1}$, respectively. Seasonal average errors were realised as 2.29 mm day$^{-1}$, 40.90% and 2.50 mm day$^{-1}$, respectively (Table 3). As an indicator of the statistical relationship between actual and estimated daily average ET$_o$ values, R$^2$ coefficients were determined as 0.83 and 0.87 for the FAO-56 and Snyder models, respectively (Fig. 9). The model that showed the nearest performance to FAO-56 in the first year was Wahed & Snyder. The MAE, MAPE and RMSE errors calculated for this model, ranged between 0.62–0.90 mm day$^{-1}$, 9.82–20.52% and 0.72–1.05 mm day$^{-1}$, respectively.

Seasonal average errors were determined as 0.71 mm day$^{-1}$, 13.52% and 0.87 mm day$^{-1}$, respectively. The R$^2$ coefficient was obtained as 0.84 for the Wahed & Snyder model. The performances of the Cuenca, Modified Snyder and Orang models in daily average ET$_o$ estimates were lower than the FAO-56 and Wahed & Snyder in the first year. Seasonal average MAE, MAPE and RMSE errors were determined as 1.02 mm day$^{-1}$, 18.87%, 1.22 mm day$^{-1}$ for Cuenca, 1.22 mm day$^{-1}$, 22.18%, 1.44 mm day$^{-1}$ for Modified Snyder and 1.25 mm day$^{-1}$, 22.72%, 1.47 mm day$^{-1}$ for Orang, respectively (Table 3). The R$^2$ coefficients of these models were calculated as 0.87, 0.86 and 0.87, respectively (Fig. 9). In the first year, the accuracy ranking of the models from best to worst according to their performance in daily average ET$_o$ estimates was as follows. FAO-56 >Wahed & Snyder >Cuenca >Modified Snyder >Orang >Snyder. Using these models, daily average ET$_o$ values were estimated with accuracy rates of 88.19% (MAPE = 11.81%), 86.48% (MAPE = 13.52%), 81.13% (MAPE = 18.87%), 77.82% (MAPE = 22.18%), 77.28% (MAPE = 22.72%) and 59.10% (MAPE = 40.90%), respectively. The accuracy of the estimated ET$_o$ values was determined as "good" (MAPE = 10–20%) for FAO-56, Wahed & Snyder, Cuenca, and "reasonable" (MAPE = 20–50%) for other models.

The daily average ET$_o$ values with the lowest and highest errors in the second year were estimated using the models of Wahed & Snyder and Snyder, respectively. The MAE, MAPE and RMSE errors determined for the Wahed & Snyder, which has the best-estimating performance, varied between 0.56–1.03 mm day$^{-1}$, 10.11–19.14% and 0.75–1.22 mm day$^{-1}$, respectively. The same errors varied between 1.32–2.20 mm day$^{-1}$, 26.95–45.81% and 1.53–2.58 mm day$^{-1}$, respectively, for the Snyder, which has the worst estimating performance. Seasonal average errors were obtained as 0.84 mm day$^{-1}$, 15.28%, 1.06 mm day$^{-1}$ for Wahed & Snyder and as 1.88 mm day$^{-1}$, 34.16%, 2.22 mm day$^{-1}$ for Snyder (Table 3). As an indicator of the statistical relationship between actual and estimated daily average ET$_o$ values, R$^2$ coefficients were determined as 0.77 and 0.76 for the Wahed & Snyder and Snyder models, respectively (Fig. 10). The FAO-56 model, which had the best estimating performance in the first year, was the model nearest in performance to Wahed & Snyder in the second year. The MAE, MAPE and RMSE errors calculated for this model, ranged between 0.60–1.15 mm day$^{-1}$, 13.33–19.68% and 0.82–1.50 mm day$^{-1}$, respectively. Seasonal average errors were determined as 0.93 mm day$^{-1}$, 16.28% and 1.20 mm day$^{-1}$, respectively. The R$^2$ coefficient was obtained as 0.72 for the FAO-56 model. The performances of the Cuenca, Modified Snyder and Orang models in daily average ET$_o$ estimates were lower than the Wahed & Snyder and FAO-56 in the second year. Seasonal average MAE, MAPE and RMSE errors were determined as 0.99 mm day$^{-1}$, 18.46%, 1.24 mm day$^{-1}$ for Cuenca, 1.07 mm day$^{-1}$, 20.07%, 1.36 mm day$^{-1}$ for Modified Snyder and 1.09 mm day$^{-1}$, 20.45%, 1.38 mm day$^{-1}$ for Orang, respectively (Table 3). The R$^2$ coefficients of these models were calculated as 0.76, 0.77 and 0.77, respectively (Fig. 10). In the second year, the accuracy ranking of the models from best to worst according to their performance in daily average ET$_o$ estimates was as follows. Wahed & Snyder >FAO-56 >Cuenca >Modified Snyder >Orang >Snyder. Using these models, daily average ET$_o$ values were estimated with accuracy rates of 84.72% (MAPE = 15.28%), 83.72% (MAPE = 16.28%), 81.54% (MAPE = 18.46%), 79.93% (MAPE = 20.07%), 79.55% (MAPE =

**Table 3 Performances of the $K_p$ models in estimating daily average $ET_o$.**

**Cuenca**

| Month | July | | August | | September | | October | | Average | |
|---|---|---|---|---|---|---|---|---|---|---|
| Year | 2020 | 2021 | 2020 | 2021 | 2020 | 2021 | 2020 | 2021 | 2020 | 2021 |
| MAE (mm day$^{-1}$) | 1.31 | 0.97 | 1.10 | 1.08 | 0.88 | 1.18 | 0.80 | 0.72 | 1.02 | 0.99 |
| MAPE (%) | 17.52 | 11.65 | 15.32 | 15.66 | 17.10 | 21.32 | 25.55 | 25.20 | 18.87 | 18.46 |
| RMSE (mm day$^{-1}$) | 1.49 | 1.17 | 1.25 | 1.35 | 1.13 | 1.47 | 0.93 | 0.91 | 1.22 | 1.24 |

**Snyder**

| Month | July | | August | | September | | October | | Average | |
|---|---|---|---|---|---|---|---|---|---|---|
| Year | 2020 | 2021 | 2020 | 2021 | 2020 | 2021 | 2020 | 2021 | 2020 | 2021 |
| **MAE (mm day$^{-1}$)** | 2.74 | 2.20 | 2.81 | 2.19 | 2.09 | 1.82 | 1.53 | 1.32 | 2.29 | 1.88 |
| MAPE (%) | 36.42 | 26.95 | 38.85 | 31.18 | 39.24 | 32.70 | 49.06 | 45.81 | 40.90 | 34.16 |
| RMSE (mm day$^{-1}$) | 2.99 | 2.50 | 2.96 | 2.58 | 2.33 | 2.28 | 1.73 | 1.53 | 2.50 | 2.22 |

**Wahed & Snyder**

| Month | July | | August | | September | | October | | Average | |
|---|---|---|---|---|---|---|---|---|---|---|
| Year | 2020 | 2021 | 2020 | 2021 | 2020 | 2021 | 2020 | 2021 | 2020 | 2021 |
| MAE (mm day$^{-1}$) | 0.90 | 0.86 | 0.71 | 0.91 | 0.62 | 1.03 | 0.64 | 0.56 | 0.71 | 0.84 |
| MAPE (%) | 11.86 | 10.11 | 9.82 | 13.01 | 11.89 | 18.86 | 20.52 | 19.14 | 13.52 | 15.28 |
| RMSE (mm day$^{-1}$) | 1.05 | 1.06 | 0.84 | 1.10 | 0.85 | 1.22 | 0.72 | 0.75 | 0.87 | 1.06 |

**FAO-56**

| Month | July | | August | | September | | October | | Average | |
|---|---|---|---|---|---|---|---|---|---|---|
| Year | 2020 | 2021 | 2020 | 2021 | 2020 | 2021 | 2020 | 2021 | 2020 | 2021 |
| MAE (mm day$^{-1}$) | 0.68 | 1.15 | 0.66 | 0.95 | 0.56 | 1.02 | 0.56 | 0.60 | 0.62 | 0.93 |
| MAPE (%) | 8.96 | 13.47 | 8.79 | 13.33 | 10.73 | 18.60 | 18.78 | 19.68 | 11.81 | 16.28 |
| RMSE (mm day$^{-1}$) | 0.93 | 1.50 | 0.81 | 1.20 | 0.73 | 1.19 | 0.66 | 0.82 | 0.79 | 1.20 |

**Modified Snyder**

| Month | July | | August | | September | | October | | Average | |
|---|---|---|---|---|---|---|---|---|---|---|
| Year | 2020 | 2021 | 2020 | 2021 | 2020 | 2021 | 2020 | 2021 | 2020 | 2021 |
| MAE (mm day$^{-1}$) | 1.52 | 1.05 | 1.39 | 1.22 | 1.09 | 1.24 | 0.88 | 0.78 | 1.22 | 1.07 |
| MAPE (%) | 20.22 | 12.77 | 19.40 | 17.60 | 20.87 | 22.31 | 28.23 | 27.60 | 22.18 | 20.07 |
| RMSE (mm day$^{-1}$) | 1.71 | 1.28 | 1.56 | 1.51 | 1.35 | 1.575 | 1.06 | 0.98 | 1.44 | 1.36 |

**Orang**

| Month | July | | August | | September | | October | | Average | |
|---|---|---|---|---|---|---|---|---|---|---|
| Year | 2020 | 2021 | 2020 | 2021 | 2020 | 2021 | 2020 | 2021 | 2020 | 2021 |
| MAE (mm day$^{-1}$) | 1.55 | 1.08 | 1.43 | 1.24 | 1.12 | 1.26 | 0.90 | 0.80 | 1.25 | 1.09 |
| MAPE (%) | 20.73 | 13.09 | 19.95 | 17.98 | 21.44 | 22.60 | 28.74 | 28.15 | 22.72 | 20.45 |
| RMSE (mm day$^{-1}$) | 1.75 | 1.31 | 1.60 | 1.54 | 1.38 | 1.60 | 1.07 | 0.99 | 1.47 | 1.38 |

**Notes.**
Mean absolute error (MAE), mean absolute percentage error (MAPE) and root mean square error (RMSE) express the deviation between the daily average actual $ET_o$ values calculated using the FAO-56 PM equation and the daily average $ET_o$ values estimated using the Cuenca, Snyder, Wahed & Snyder, FAO-56, Modified Snyder, and Orang models.

20.45%) and 65.84% (MAPE = 34.16%), respectively. The accuracy of the estimated $ET_o$ values was determined as "good" (MAPE = 10–20%) for Wahed & Snyder, FAO-56 Cuenca, and "reasonable" (MAPE = 20–50%) for other models. Considering the results obtained for both years, it has been seen that the nearest values to the daily average actual $ET_o$ values can be estimated in Kahramanmaraş conditions using the models of FAO-56
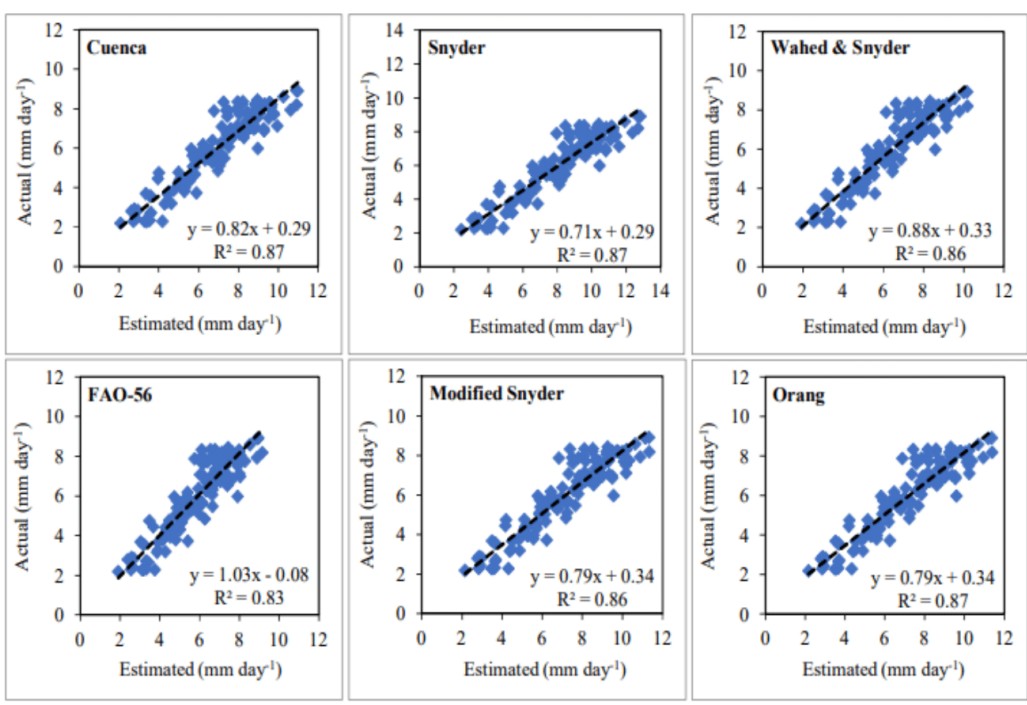

**Figure 9** Statistical analysis of the relationship between actual and estimated daily average reference evapotranspiration ($ET_o$) values (2020).

and Wahed & Snyder, which have similar performances. The best alternative of these models was Cuenca. Wahed & Snyder outperformed the FAO-56 in July and August, while exhibiting underperformed the FAO-56 in September and October. Wahed & Snyder, Cuenca, Modified Snyder, Orang and Snyder models overestimated daily average $ET_o$ values by 14.40%, 18.67%, 21.13%, 21.59% and 37.53%, respectively, while FAO-56 model underestimated by 14.05%. The Modified Snyder and Orang models showed very similar performances in both years.

The monthly average actual $ET_o$ values were determined as 7.62 mm day$^{-1}$, 7.35 mm day$^{-1}$, 5.40 mm day$^{-1}$ and 3.27 mm day$^{-1}$ for the months of July, August, September and October in the first year, respectively. The same values were obtained as 8.27 mm day$^{-1}$, 7.08 mm day$^{-1}$, 5.55 mm day$^{-1}$ and 3.20 mm day$^{-1}$ for the second year, respectively. The nearest values to the monthly average actual $ET_o$ values were estimated by the FAO-56 model (7.33 mm day$^{-1}$, 6.98 mm day$^{-1}$, 5.43 mm day$^{-1}$, 3.56 mm day$^{-1}$) in the first year and by the Wahed & Snyder model (8.00 mm day$^{-1}$, 7.16 mm day$^{-1}$, 5.70 mm day$^{-1}$, 3.44 mm day$^{-1}$) in the second year, similar to the daily average $ET_o$ estimates. The furthest values to the monthly average actual $ET_o$ values were obtained with the Snyder model in both years as in the daily average $ET_o$ estimates (Table 4).

The MAE, MAPE, RMSE errors and $R^2$ coefficients for the FAO-56 and Wahed & Snyder models, which have the best-estimating performances in monthly average $ET_o$ estimates, were determined as 0.25 mm day$^{-1}$, 4.57%, 0.28 mm day$^{-1}$, 0.99 and 0.44 mm day$^{-1}$, 8.11%, 0.44 mm day$^{-1}$, 0.99 in the first year, respectively. For the same models, 0.51 mm

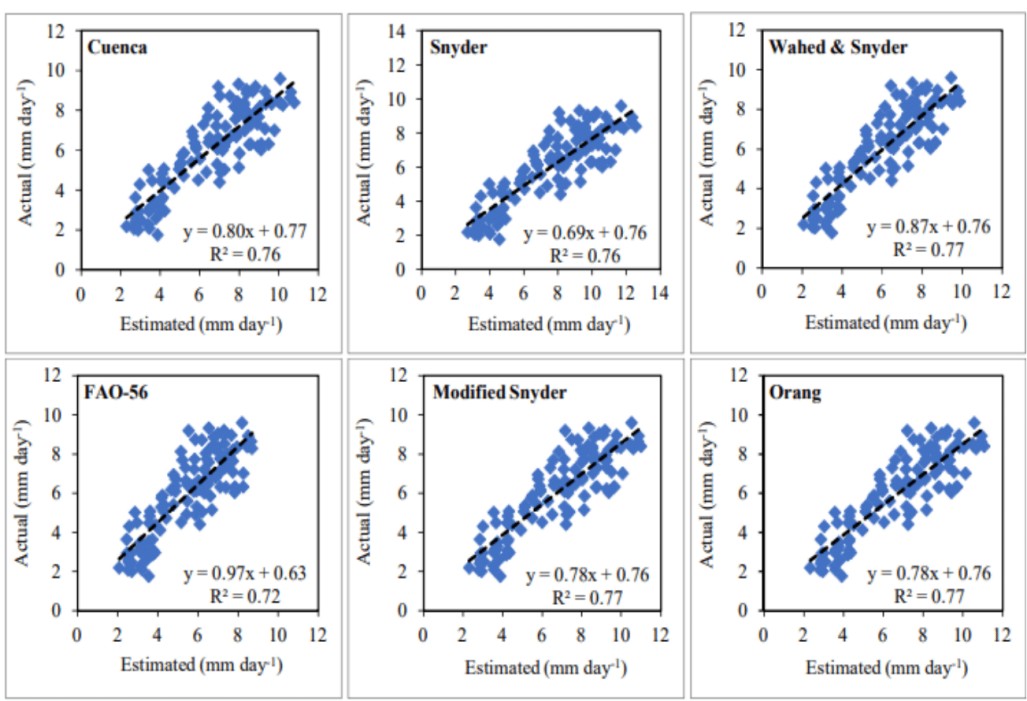

**Figure 10** Statistical analysis of the relationship between actual and estimated daily average reference evapotranspiration ($ET_o$) values (2021).

**Table 4  Monthly averages of the actual and estimated daily $ET_o$ (mm day$^{-1}$) values.**

| Model/Month (2020) | July | August | September | October | Average |
|---|---|---|---|---|---|
| Actual | 7.62 | 7.35 | 5.40 | 3.27 | 5.91 |
| Cuenca | 8.73 | 8.40 | 6.23 | 3.96 | 6.83 |
| Snyder | 10.17 | 9.78 | 7.26 | 4.61 | 7.96 |
| Wahed & Snyder | 8.06 | 7.83 | 5.81 | 3.68 | 6.35 |
| FAO-56 | 7.33 | 6.98 | 5.43 | 3.56 | 5.83 |
| Modified Snyder | 8.97 | 8.72 | 6.46 | 4.09 | 7.07 |
| Orang | 9.02 | 8.77 | 6.50 | 4.11 | 7.10 |
| **Model/Month (2021)** | **July** | **August** | **September** | **October** | **Average** |
| Actual | 8.27 | 7.08 | 5.55 | 3.20 | 6.03 |
| Cuenca | 8.61 | 7.72 | 6.17 | 3.71 | 6.56 |
| Snyder | 10.02 | 8.98 | 7.19 | 4.32 | 7.63 |
| Wahed & Snyder | 8.00 | 7.16 | 5.70 | 3.44 | 6.08 |
| FAO-56 | 7.17 | 6.58 | 5.22 | 3.30 | 5.57 |
| Modified Snyder | 8.91 | 7.98 | 6.34 | 3.83 | 6.77 |
| Orang | 8.95 | 8.02 | 6.38 | 3.85 | 6.80 |

day$^{-1}$, 7.36%, 0.40 mm day$^{-1}$, 0.99 and 0.19 mm day$^{-1}$, 3.65%, 0.20 mm day$^{-1}$, 0.99 values were obtained in the second year, respectively. The MAE, MAPE, RMSE errors and $R^2$ coefficients of the Snyder model, which has the worst estimating performances

in monthly $ET_o$ estimates, were determined as 2.05 mm day$^{-1}$, 35.49%, 2.10 mm day$^{-1}$, 0.99 in the first year and 1.60 mm day$^{-1}$, 28.14%, 1.63 mm day$^{-1}$, 0.99 in the second year, respectively (Table 5). The performance of the FAO-56, Wahed & Snyder, Cuenca, Modified Snyder, Orang and Snyder models in monthly average $ET_o$ estimates were 9.43%, 10.00%, 6.72%, 5.28%, 5.01% and 9.15% higher than their performance in daily average $ET_o$ estimates, respectively. The accuracy of the estimated monthly $ET_o$ values was determined as "excellent" (MAPE<10%) for Wahed & Snyder, FAO-56, and "good" (MAPE = 10–20%) for Cuenca, Modified Snyder, Orang, and "reasonable" (MAPE = 20–50%) for Snyder.

The monthly total actual $ET_o$ values varied between 101.22–236.26 mm and 99.13–256.43 mm for the July–October periods of 2020 and 2021, respectively. The monthly total $ET_o$ values estimated for the first year using FAO-56, Wahed & Snyder, Cuenca, Modified Snyder, Orang and Snyder models ranged between 110.21–227.35 mm, 113.91–249.87 mm, 122.61–270.71 mm, 126.81–278.14 mm, 127.49–279.70 mm and 142.80–315.35 mm, respectively. Using the same models, the estimated values for the second year varied between 102.39–222.25 mm, 106.61–248.01 mm, 114.93–267.03 mm, 118.68–276.10 mm, 119.32–277.56 mm and 133.95–310.71 mm, respectively (Fig. 11). The seasonal total actual $ET_o$ values were realised as 727.38 mm and 741.48 mm for both years, respectively. The seasonal total $ET_o$ values estimated for the first year using FAO-56, Wahed & Snyder, Cuenca, Modified Snyder, Orang and Snyder models were determined as 716.80 mm, 780.75 mm, 840.49 mm, 869.14 mm, 873.81 mm and 979.03 mm, respectively. The estimated values for the second year were obtained as 685.36 mm, 747.64 mm, 806.33 mm, 822.28 mm, 836.79 mm and 938.75 mm, respectively. The nearest values to the monthly and seasonal total actual $ET_o$ values were obtained with FAO-56 and Wahed & Snyder models. The furthest values were estimated with the Snyder model.

# DISCUSSION

In this study conducted during the July–October periods of 2020 and 2021, the daily climate data and daily total $E_{pan}$ values measured with the PLC-controlled sensors were used. The two-year averages of the monthly average $ET_o$ values determined for July, August, September and October were obtained as 7.95 mm day$^{-1}$, 7.22 mm day$^{-1}$, 5.48 mm day$^{-1}$ and 3.24 mm day$^{-1}$, respectively. In similar studies conducted for Kahramanmaraş using the FAO-56 PM equation; 7.13 mm day$^{-1}$, 6.38 mm day$^{-1}$, 4.50 mm day$^{-1}$, 2.38 mm day$^{-1}$ (TAGEM, 2017), 6.65 mm day$^{-1}$, 6.15 mm day$^{-1}$, 4.82 mm day$^{-1}$, 2.46 mm day$^{-1}$ (Gençoğlan et al., 2019) and 7.00 mm day$^{-1}$, 6.57 mm day$^{-1}$, 4.13 mm day$^{-1}$, 2.96 mm day$^{-1}$ (Kaymaz, 2020) values were obtained for the same months, respectively. The daily $E_{pan}$ values measured from the PLC-controlled class-A pan evaporimeter during the July–October periods of both years, varied between 3.00–16.00 mm day$^{-1}$. The two-year averages of the monthly total $E_{pan}$ values determined for July, August, September and October were obtained as 385.00 mm, 362.50 mm, 267.50 mm and 171 mm, respectively. The two-year average seasonal total $E_{pan}$ was determined as 1,186 mm. In similar studies realised for Kahramanmaraş; daily total $E_{pan}$ values ranging from 2.30 mm day$^{-1}$ to 18.20 mm day$^{-1}$ and seasonal total $E_{pan}$ values reaching 1,104 mm were measured for the July–October

**Table 5  Performances of the $K_p$ models in estimating monthly average $ET_o$.**

| Cuenca | | | | | | | | | | |
| --- | --- | --- | --- | --- | --- | --- | --- | --- | --- | --- |
| Month | July | | August | | September | | October | | Average | |
| Year | 2020 | 2021 | 2020 | 2021 | 2020 | 2021 | 2020 | 2021 | 2020 | 2021 |
| MAE (mm day$^{-1}$) | 1.11 | 0.34 | 1.05 | 0.64 | 0.83 | 0.62 | 0.69 | 0.55 | 0.92 | 0.63 |
| MAPE (%) | 14.57 | 4.11 | 14.29 | 9.04 | 15.37 | 11.17 | 21.10 | 15.94 | 16.33 | 10.07 |
| RMSE (mm day$^{-1}$) | 1.15 | 0.66 | 0.96 | 0.55 | 0.87 | 0.50 | 0.72 | 0.41 | 0.94 | 0.54 |

| Snyder | | | | | | | | | | |
| --- | --- | --- | --- | --- | --- | --- | --- | --- | --- | --- |
| Month | July | | August | | September | | October | | Average | |
| Year | 2020 | 2021 | 2020 | 2021 | 2020 | 2021 | 2020 | 2021 | 2020 | 2021 |
| MAE (mm day$^{-1}$) | 2.55 | 1.75 | 2.43 | 1.90 | 1.86 | 1.64 | 1.34 | 1.12 | 2.05 | 1.60 |
| MAPE (%) | 33.47 | 21.16 | 33.06 | 26.83 | 34.44 | 29.55 | 40.98 | 35.00 | 35.49 | 28.14 |
| RMSE (mm day$^{-1}$) | 2.51 | 1.84 | 2.49 | 1.89 | 1.96 | 1.67 | 1.45 | 1.12 | 2.10 | 1.63 |

| Wahed & Snyder | | | | | | | | | | |
| --- | --- | --- | --- | --- | --- | --- | --- | --- | --- | --- |
| Month | July | | August | | September | | October | | Average | |
| Year | 2020 | 2021 | 2020 | 2021 | 2020 | 2021 | 2020 | 2021 | 2020 | 2021 |
| MAE (mm day$^{-1}$) | 0.44 | 0.27 | 0.48 | 0.08 | 0.41 | 0.15 | 0.41 | 0.24 | 0.44 | 0.19 |
| MAPE (%) | 5.77 | 3.27 | 6.53 | 1.13 | 7.59 | 2.70 | 12.54 | 7.50 | 8.11 | 3.65 |
| RMSE (mm day$^{-1}$) | 0.53 | 0.20 | 0.43 | 0.21 | 0.43 | 0.23 | 0.36 | 0.14 | 0.44 | 0.20 |

| FAO-56 | | | | | | | | | | |
| --- | --- | --- | --- | --- | --- | --- | --- | --- | --- | --- |
| Month | July | | August | | September | | October | | Average | |
| Year | 2020 | 2021 | 2020 | 2021 | 2020 | 2021 | 2020 | 2021 | 2020 | 2021 |
| MAE (mm day$^{-1}$) | 0.29 | 1.10 | 0.37 | 0.50 | 0.03 | 0.33 | 0.29 | 0.10 | 0.25 | 0.51 |
| MAPE (%) | 3.81 | 13.30 | 5.03 | 7.06 | 0.56 | 5.95 | 8.87 | 3.13 | 4.57 | 7.36 |
| RMSE (mm day$^{-1}$) | 0.33 | 0.50 | 0.29 | 0.40 | 0.26 | 0.40 | 0.23 | 0.27 | 0.28 | 0.40 |

| Modified Snyder | | | | | | | | | | |
| --- | --- | --- | --- | --- | --- | --- | --- | --- | --- | --- |
| Month | July | | August | | September | | October | | Average | |
| Year | 2020 | 2021 | 2020 | 2021 | 2020 | 2021 | 2020 | 2021 | 2020 | 2021 |
| MAE (mm day$^{-1}$) | 1.35 | 0.64 | 1.37 | 0.90 | 1.06 | 0.79 | 0.82 | 0.63 | 1.15 | 0.74 |
| MAPE (%) | 17.72 | 7.74 | 18.64 | 12.71 | 19.63 | 14.23 | 25.08 | 19.69 | 20.27 | 13.59 |
| RMSE (mm day$^{-1}$) | 1.39 | 0.94 | 1.27 | 0.86 | 1.10 | 0.75 | 0.86 | 0.59 | 1.16 | 0.78 |

| Orang | | | | | | | | | | |
| --- | --- | --- | --- | --- | --- | --- | --- | --- | --- | --- |
| Month | July | | August | | September | | October | | Average | |
| Year | 2020 | 2021 | 2020 | 2021 | 2020 | 2021 | 2020 | 2021 | 2020 | 2021 |
| MAE (mm day$^{-1}$) | 1.40 | 0.68 | 1.42 | 0.94 | 1.10 | 0.83 | 0.84 | 0.65 | 1.19 | 0.78 |
| MAPE (%) | 18.37 | 8.22 | 19.32 | 13.28 | 20.37 | 14.96 | 25.69 | 20.31 | 20.94 | 14.19 |
| RMSE (mm day$^{-1}$) | 1.44 | 0.98 | 1.32 | 0.91 | 1.14 | 0.78 | 0.88 | 0.61 | 1.19 | 0.81 |

**Notes.**
Mean absolute error (MAE) and mean absolute percentage error (MAPE) express the deviation between the monthly average actual $ET_o$ values calculated using the FAO-56 PM equation and the monthly average $ET_o$ values estimated using the Cuenca, Snyder, Wahed & Snyder, FAO-56, Modified Snyder, and Orang models.

period (*Gençoğlan & Gençoğlan, 2018*; *Diş, 2023*). Moreover, the four-year averages of the monthly total $E_{pan}$ values measured at the Kahramanmaraş ground-based meteorological observation station for July, August, September and October of 2018, 2019, 2020 and 2021 were determined as 376 mm, 324 mm, 240 mm and 156 mm, respectively (*Turkish State Meteorological Service, 2022*). Although there were minor differences between the values

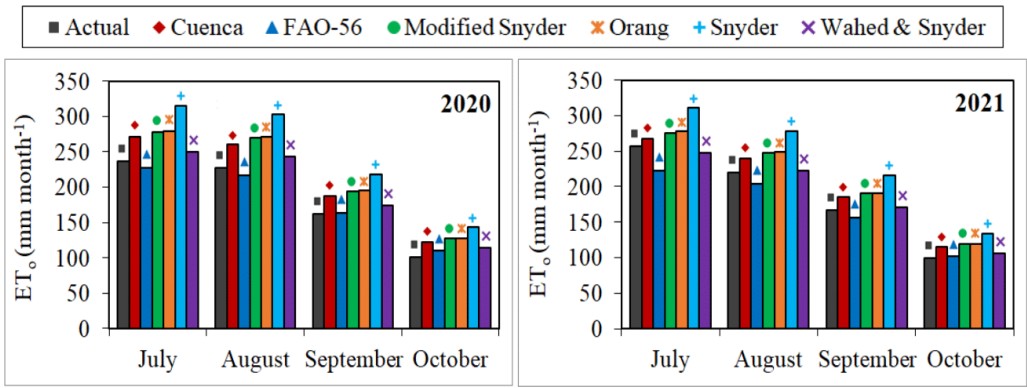

**Figure 11  Monthly total actual and estimated reference evapotranspiration (ET$_o$) values.** Each bar on the graphs represents the monthly total actual and estimated ET$_o$ values for the July–October periods of 2020 and 2021.

determined within the scope of these studies and the values obtained using PLC-controlled sensors, they generally exhibited similarities. These differences are thought to arise from time-dependent changes in air temperature, relative humidity, wind velocity, and solar radiation parameters. This result revealed that the daily climate data and daily E$_{pan}$ values measured using PLC-controlled sensors have an acceptable level of consistency and reliability. It was observed that the PLC-controlled sensors used in this two-year study reduced labour and time usage. Additionally, many researchers have reported that innovative automation control devices such as PLC increase the accuracy of measurement processes and minimize human-caused measurement errors (*Işık et al., 2017*; *Mantri et al., 2018*; *Öter & Bahar, 2018*).

*Irmak & Haman (2003)* and *Gundekar et al. (2008)* stated that RMSE errors less than 0.50 mm day$^{-1}$ were considered acceptable for ET$_o$ values estimated using different estimation methods. In this study conducted using two-year data set measured with PLC-controlled sensors, the ET$_o$ values with the highest accuracy were estimated by the Wahed & Snyder and FAO-56 models. The average RMSE errors of the daily and monthly average ET$_o$ values estimated by these models were determined as 0.97 mm day$^{-1}$, 1.00 mm day$^{-1}$ and 0.32 mm day$^{-1}$, 0.34 mm day$^{-1}$, respectively. Although these RMSE errors were slightly above 0.50 mm day$^{-1}$ for daily estimates, they were below 0.50 mm day$^{-1}$ for monthly estimates. The RMSE errors of the daily and monthly average ET$_o$ values estimated by other K$_p$ coefficient estimation models evaluated within the scope of the study varied between 1.23–2.36 mm day$^{-1}$ and 0.75–1.87 mm day$^{-1}$, respectively. According to this evaluation based on RMSE, It has been concluded that none of the six K$_p$ models can be used to estimate the daily average ET$_o$ in Kahramanmaraş, and only Wahed & Snyder and FAO-56 can be used to estimate the monthly average ET$_o$ without calibration.

*Gundekar et al. (2008)*, *Sabziparvar et al. (2010)*; *Pradhan et al. (2013)*, *Kaya et al. (2012)*; *Aydın (2019)* and *Tya, Sunday & Vanke (2020)* reported that Snyder is the model with the best-estimating performance in semi-arid climate conditions. Similarly *Irmak, Haman & Jones (2002)*; *SreeMaheswari & Jyothy (2017)*; *Tabari, Grismer & Trajkovic*

*(2013)*; *Kar et al. (2017)*; *Khobragade et al. (2019)* and *Mahmud et al. (2020)* stated that Snyder and Cuenca are the models with the best-estimating performance in humid climatic conditions. The Snyder model, which generally has the best-estimating performance in semi-arid and humid climatic conditions, showed the worst performance (MAE = 1.83 mm day$^{-1}$, MAPE = 31.82%, RMSE = 1.87 mm day$^{-1}$) in this study conducted in Kahramanmaraş which has a semi-arid Mediterranean climate. The accuracy ranking of the six pan coefficient estimation models considered in this study, where Wahed & Snyder (MAE = 0.32 mm day$^{-1}$, MAPE = 5.88%, RMSE = 0.32 mm day$^{-1}$) and FAO-56 (MAE = 0.38 mm day$^{-1}$, MAPE = 5.97%, RMSE = 0.34 mm day$^{-1}$) models have the best-estimating performance, was as follows. Wahed & Snyder>FAO-56>Cuenca>Modified Snyder>Orang>Snyder. Similarly *Aschonitis, Antonopoulos & Papamichail (2012)* declared that the models with the best and worst estimating performances were Cuenca (MAE = 0.14 mm day$^{-1}$, RMSE = 0.61 mm day$^{-1}$) and Snyder (MAE = 2.53 mm day$^{-1}$, RMSE = 2.73 mm day$^{-1}$), respectively, in their study conducted in the Thessaloniki plain of Greece, where has a semi-arid Mediterranean climate. The accuracy ranking of the seven models discussed in this study, in which Wahed & Snyder and FAO-56 models were not evaluated, was as follows. Cuenca >Raghuwanshi & Wallender>Allen & Pruitt>Pereira>Orang >Snyder. In another study conducted in Mediterranean climate conditions, *Koç (2022)* reported that Wahed & Snyder was the best performing model (MAE = 0.43 mm day$^{-1}$, RMSE = 0.55 mm day$^{-1}$) and Orang was the worst performing model (MAE = 1.81 mm day$^{-1}$, RMSE = 1.87 mm day$^{-1}$) in Adana, 195 km from Kahramanmaraş. The accuracy ranking of the eight models discussed in this study, was as follows. Wahed & Snyder>Modified Snyder>Cuenca>Raghuwanshi & Wallender>Pereira>Allen & Pruitt>Snyder>Orang. Using the Wahed & Snyder model in Adana conditions, monthly average $K_p$ coefficients were estimated as 0.65, 0.65, 0.64 and 0.63 for the months of July, August, September and October, respectively. Similarly, using the same model, the $K_p$ coefficients of 0.65, 0.64, 0.64 and 0.65 were obtained for the same months in Kahramanmaraş conditions.

# CONCLUSIONS

In this study, conducted in Kahramanmaraş, which has a semi-arid Mediterranean climate in Turkey during the July–October periods of 2020 and 2021, the usability levels of six $K_p$ models in estimating daily and monthly average $ET_o$ were evaluated. The daily average $ET_o$ values were estimated on a model basis by multiplying the $K_p$ coefficients with the daily $E_{pan}$ values. The daily $E_{pan}$ values were measured using an ultrasonic sensor sensitive to water level. The $ET_o$ values determined with the FAO-56 PM equation were accepted as actual values. The $ET_o$ values estimated by the $K_p$ estimation models were compared with the actual $ET_o$ values, and their usability levels were revealed. The Wahed & Snyder model outperformed the other models in estimating daily (MAE = 0.78 mm day$^{-1}$, MAPE = 14.40%, RMSE = 0.97 mm day$^{-1}$, $R^2 = 0.82$) and monthly (MAE = 0.32 mm day$^{-1}$, MAPE = 5.88%, RMSE = 0.32 mm day$^{-1}$, $R^2 = 0.99$) average $ET_o$. FAO-56 was the model that performed nearest to Wahed & Snyder. The Snyder model presented the worst performance in estimating daily (MAE = 2.09 mm day$^{-1}$, MAPE = 37.53%, RMSE = 2.36

mm day$^{-1}$, $R^2 = 0.82$) and monthly (MAE $= 1.83$ mm day$^{-1}$, MAPE $= 31.82\%$, RMSE $= 1.87$ mm day$^{-1}$, $R^2 = 0.99$) average ET$_o$. It has been concluded that none of the six K$_p$ models can be used to estimate the daily ET$_o$ in Kahramanmaraş, and only Wahed & Snyder and FAO-56 can be used to estimate the monthly ET$_o$ without calibration.

### Funding
The author received no funding for this work.

### Competing Interests
The author declares that he has no competing interests.

### Author Contributions
- Selçuk Usta conceived and designed the experiments, performed the experiments, analyzed the data, prepared figures and/or tables, authored or reviewed drafts of the article, and approved the final draft.

### Data Availability
Raw data are available in the Supplemental Files.

### Supplemental Information
Supplemental information for this article can be found online at http://dx.doi.org/10.7717/peerj.17685#supplemental-information.

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
