# Peer review of "Estimation of reference evapotranspiration using some class-A pan evaporimeter pan coefficient estimation models in Mediterranean–Southeastern Anatolian transitional zone conditions of Turkey"

_PeerJ, doi:10.7717/peerj.17685_

## Round 0.1 · original submission · Major Revisions

Dear Dr Usta,

Two independent experts have agreed to evaluate your work. Both were in consensus that this work may be published in the PeerJ journal, but it needs thorough revisions beforehand. Please review the detailed comments from the reviewers and address each of them accordingly.

With best regards,

·

Basic reporting

The introduction and results sections of the manuscript need major improvements. Specific suggestion are given below.

Introduction: The justification or rationale and objectives of the study should be incorporated into the introduction to provide context and clarity about the research aims.

Results: The results section should be rewritten to focus on the key findings of the study, presenting them clearly and concisely.

Discussion: The discussion part of the manuscript may be strengthened by comparing and contrasting the findings with those of more relevant manuscripts, which can help provide a deeper understanding of the research outcomes and their implications.

The other sections of the manuscript have some grammatical and formatting issues, which are mentioned as comments to the specific lines (please the attached annotated PDF file).

Experimental design

No comment

Validity of the findings

The findings of this study have positive implications. However, it would be better to conclude about the performance of the models based on the more statistical parameters rather than depending on the only one.

Additional comments

Try to use high-resolution pictures.

Reviewer 2 ·

Basic reporting

The submitted paper is found to be interesing and valuable to a number of readers. The research topic is an up-to-date attempt to compare different methods for the estimation of evapotranspiration , basing on one measuring, meteorological station and direct some class-A pan evaporometer measurements. Such organization of the research is considered to be most proper, however, direct hypothesis is missing. Is the comparison of the methods the only goal of the paper? Should the authors kindly state, what could be the real achievement or novelty through comparing different methods for one region solely? The structure of the paper is appropriate and the literature references are sufficient.

Experimental design

The presented knowledge seems to bridge the existing gap, nonetheless, only for one climatic region. The undertaken analyes fall within the scope of the journal. The research question, to the best of my belief, and respectful of the tremendous work of the authors, is of secondary importance. The investigation methods are sufficiently described, however, according to the conclusions, the obtained ET differences between the metohds are statistically insignificant, that makes them not vital in fact. I would rather compare the existing methods ( Cuenca, 289 FAO-56, Modified Snyder, Orang, Snyder and Wahed & Snyder ) for a couple of meteorological stations, to see which differences may occur, since we take a few regions and apply a number of methods to one station in a region. If available, pan evaporometer measurements could be also compared with FAO method for a number of regions. I strongly suggest, the authors try to take the data of the meteorological stations from few regions of Turkey, and apply different ET calcualtion methods to one station. Would you furhter state, if different methods for one station exert statistically significant differences between results ? So far the conclusions are quite plain and superficial. What could be the differences in evapotranspiration by FAO method for particular months in different regions of the country? (Turkey)

Validity of the findings

The research question is well establised but it may not be based on one station only. As far as I am concerned, it would be better to compare the ET results between different climatic regions of the country or continent, at least as an introductory part.

---

## Round 0.2 · accepted · Accept

Dear Dr. Usta,
The reviewer has accepted all the changes you made, and your work can be published in PeerJ in its current version - congratulations!

·

Basic reporting

Authors did a great job. The article in the present state has properly addressed all the comments. The quality has been improved enormously.

Experimental design

It's perfectly alright.

Validity of the findings

The findings are valuable in the current state of knowledege of this research area.

Additional comments

The article can be accepted for publication in the said journal.